# Association between the Use of Antibiotics and the Development of Acute Renal Injury in Patients Hospitalized for COVID-19 in a Hospital in the Peruvian Amazon

**DOI:** 10.3390/jcm11154493

**Published:** 2022-08-02

**Authors:** Luccio Romaní, Darwin A. León-Figueroa, David Rafael-Navarro, Joshuan J. Barboza, Alfonso J. Rodriguez-Morales

**Affiliations:** 1Facultad de Medicina Humana, Universidad San Martín de Porres, Chiclayo 15011, Peru; luccioromani@gmail.com (L.R.); darwin_leon@usmp.pe (D.A.L.-F.); 2Emerge, Unidad de Investigación en Enfermedades Emergentes y Cambio Climático, Facultad de Salud Pública y Administración, Universidad Peruana Cayetano Heredia, Lima 15102, Peru; 3Facultad de Medicina Humana, Universidad Nacional de Ucayali, Pucallpa 25004, Peru; dafal2608@gmail.com; 4Vicerrectorado de Investigación, Universidad Norbert Wiener, Lima 15046, Peru; 5Grupo de Investigación Biomedicina, Faculty of Medicine, Fundación Universitaria Autónoma de las Américas, Pereira 660003, Risaralda, Colombia; alfonso.rodriguez@uam.edu.co or; 6Institución Universitaria Visión de las Américas, Pereira 660003, Risaralda, Colombia; 7Master of Clinical Epidemiology and Biostatistics, Universidad Científica del Sur, Lima 15067, Peru

**Keywords:** COVID-19, acute kidney injury, intensive care units, antibacterials

## Abstract

*Introduction:* A significant antibiotic prescribing pattern associated with the COVID-19 pandemic has been described. Multiple protocols included empirical antimicrobials, leading to a substantial increase in antimicrobial consumption in medical care. A higher mortality rate is described among patients diagnosed with COVID-19 who received antibiotics. *Objectives:* To determine the association between the use of antibiotics and the development of acute renal injury in patients infected with SARS-CoV-2 in patients treated at the Hospital II EsSalud de Ucayali, 2021. *Methods:* A cross-sectional-analytical study was conducted, evaluating the medical records of patients admitted to the intensive care unit between July 2020 and July 2021. For the statistical analysis, measures of central tendency and dispersion, statistical hypothesis contrast tests were used in relation to acute kidney injury (AKI), antibiotic use and associated factors, derived from linear regression models. *Results:* The factors that were positively associated with the development of AKI were sepsis (aPR: 2.86; 95% CI: 1.26–6.43), shock (aPR:2.49; 95% CI: 1.28–4.86), mechanical ventilation (aPR:9.11; 95% CI: 1.23–67.57), and use of vancomycin (aPR: 3.15; 95% CI: 1.19–8.27). *Conclusions*: In the Peruvian Amazon, there is a high consumption and inadequate prescription of antibiotics. The drugs most commonly used for the treatment of COVID-19 were: aminoglycosides, vancomycin, ivermectin, azithromycin, tocilizumab, and corticosteroids. The development of AKI among hospitalized patients was found to be related to vancomycin administration. In addition, an association was found with the use of mechanical ventilation, a high body mass index, and the presence of complications such as sepsis or shock. Therefore, inappropriate antibiotic use for COVID-19 has been associated with multiple negative outcomes and consequences.

## 1. Introduction

Coronavirus disease 2019 (COVID-19) is a newly discovered contagious disease caused by Severe Acute Respiratory Syndrome Coronavirus–2 (SARS-CoV-2). It was first reported in Wuhan, China, and has been classified by the World Health Organization (WHO) as a pandemic [1].

The clinical features of COVID-19 vary individually, ranging from no clinical symptoms (asymptomatic) to symptoms of mild or severe respiratory disease. Common symptoms of COVID-19 include fever, fatigue, dry cough, and muscle pain, and critically ill patients may rapidly progress to acute respiratory distress syndrome (ARDS), septic shock, metabolic acidosis, coagulation dysfunction, acute renal injury, and death [2].

Acute kidney injury (AKI) is one of the most important complications of critical illness and is a major public health problem, commonly associated with sepsis, cardiac dysfunction and exposure to nephrotoxic drugs; however, less common causes of AKI (structural, immune-mediated and microvascular) can lead to devastating patient outcomes when the underlying diagnosis is overlooked or delayed [3]. AKI, formerly known as acute renal failure, refers to an acute and often reversible reduction in renal function [4]. AKI is currently defined by the Kidney Disease: Improving Global Outcomes (KDIGO) by the presence of any of the following criteria [5]: an increase in serum creatinine by 0.3 mg/dL or more (26.5 micromoles/L or more) in 48 h, an increase in serum creatinine to 1.5 times or more the initial value in the previous seven days, and a urine volume of less than 0.5 mL/kg/h for at least 6 h.

Renal involvement in SARS-CoV-2 infection occurs when SARS-CoV-2 spike proteins bind to ACE2 receptors on renal cells, acting as entry vectors for the virus. This allows the virus to induce the cytokine storm causing acute kidney injury [6]. Renal complications in SARS-CoV-2 infection may occur due to direct cytopathic effects and secondary damage as a result of the coexistence of a systemic inflammatory response or the use of renal toxic therapies, hypoxia induced by respiratory distress syndrome, or multiorgan dysfunction. Histologic findings have shown acute tubular necrosis and interstitial inflammation. Clinical manifestations of renal involvement in COVID-19 include proteinuria, hematuria, and AKI [7]. Thus, AKI is a marker of disease severity and is a negative prognostic factor for survival [8].

The incidence of AKI in patients infected with SARS-CoV-2 has been reported unevenly throughout the world (6): China reported incidences from 0.5% to 29%, Italy reported incidences from 17.9% to 72.7%, Korea from 9.2% to 18.3%, Spain from 19.7% to 69.2%, the United States from 5.8% to 56.9%, Germany from 52.2% to 74.6%, and France and Belgium from 4.7% to 55.9%. This variation could be due to differences in health care systems, indications for hospitalization, allocation of patients to levels of care, and patient admission rates that exist in all countries. It was also reported that AKI rates increased significantly from 14.5% to 50% in patients with COVID-19 admitted to intensive care units (ICU) [9,10]. A study reported that the current COVID-19 pandemic in its second- and third-wave phases has posed new challenges in different countries, generating an increase in the hospitalization of patients with high morbidity and mortality [11,12].

In patients with severe disease, the WHO recommended the provision of antimicrobial therapy to prevent further complications of infection, leading to ARDS and multiorgan failure [13]. However, the scientific literature had many gaps and uncertainties regarding the impact of antimicrobial therapy during the pandemic, particularly in ICUs.

The scientific literature has described significant antimicrobial prescribing in hospitals worldwide, which could be a temporal pattern associated with the COVID-19 pandemic. Since the first wave of this pandemic, multiple protocols included empirical antimicrobials such as ceftriaxone and azithromycin, leading to a substantial increase in antimicrobial consumption in various health care settings [13,14].

The most frequently used antibiotics in ICUs were third-generation cephalosporins (36.8%) and azithromycin (34.2%), which were the most prevalent antibiotics reported during the management of COVID-19 in patients admitted to the ICU [13]. Macrolides and hydroxychloroquine intended for the treatment of COVID-19 during the pandemic were initially used as an adjunctive anti-inflammatory and antiviral therapy rather than for their antibacterial properties. However, conflicting studies and reports delayed their removal from management protocols [15]. Despite this, the use of antimicrobial therapy is still indicated for various stages of the disease.

A recent study describes a higher mortality rate among patients diagnosed with COVID-19 who received antibiotics such as cefepime, ceftriaxone, vancomycin, and azithromycin compared to those who did not [16]. In contrast, findings from another study showed no significant difference in mortality between persons with COVID-19 who did or did not receive antimicrobial treatments [17].

As a result, the effects of antibiotics used for superimposed bacterial infections and AKI on the outcomes of patients with COVID-19 have not yet been clearly described. Given the above, the present investigation arose from the need to describe the use of antibiotics in COVID-19 and their association with the development of AKI.

## 2. Materials and Methods

### 2.1. Study Design and Study Population

A quantitative, observational, cross-sectional analytical study was carried out. The population consisted of patients who attended the intermediate care unit and the intensive care unit of the Hospital EsSalud II of Pucallpa between July 2020 and July 2021, with a diagnosis of COVID-19, confirmed through serological, antigenic, and/or molecular tests.

A non-probabilistic census-type sampling was carried out. The unit of analysis consisted of the patients’ medical records. All patients over 18 years of age who were seen during the study period and who had a complete medical history were included. Clinical histories whose data were not complete and/or were not filled out properly were excluded, making analysis impossible. All patients with chronic kidney disease, eGFR < 30 mL/min/1.73 m^2^, or on the previous hemodialysis were excluded.

### 2.2. Data Collection

The instrument consisted of a data collection form that evaluated sociodemographic characteristics such as age (years of age), sex (male/female), and place of origin, and clinical characteristics such as creatinine (mg/dL), weight (kg), height (cm), body mass index (BMI) (kg/m^2^), time of illness (days), days of hospitalization (days), comorbidities (hypertension, diabetes mellitus, neoplasms, cardiovascular disease, pulmonary disease), complications (sepsis, heart disease, lung disease, pulmonary disease), complications (sepsis, septic shock, mechanical ventilation, venous thromboembolism), death, recovery, renal replacement therapy, and drugs used, such as aminoglycosides, vancomycin, ivermectin, azithromycin, tocilizumab, and corticosteroids.

Acute kidney injury (AKI) was assessed as the deterioration of kidney function over hours to days, with an inability of the kidney to maintain fluid and electrolyte balance, and clear waste products of metabolism with stage 3 [18] according to the Kidney Disease Improving Global Outcome (KDIGO) criteria clinical practice guidelines [19].

Permission was obtained from the Hospital EsSalud II of Pucallpa for its execution, as well as from the head of the Nephrology service for access to the initial registry and later to the clinical histories of the population to be studied. Likewise, the list of COVID-19-infected patients admitted to the critical care wards was obtained from the hospital’s health intelligence office.

Once permissions were obtained in all services, the variables were collected in a data collection form hosted in the KoboToolbox^®^ platform, according to the variables indicated in the data collection form. Data were entered individually by two previously trained data collectors. Data collection was supervised by one of the authors. Subsequently, the quality of the data was checked independently by each of the authors. Data coding was performed automatically by the platform. Finally, the data were exported in a Microsoft Excel^®^ spreadsheet, which will be deleted one year after the publication of the study.

### 2.3. Statistical Analysis

Data analysis was performed using the STATA v.17.0 statistical package (StataCorp^®^). For the descriptive analysis of the categorical variables, frequencies and percentages were used for frequencies and percentages, and for quantitative variables, measures of central tendency and dispersion (numerical variables) were used for measures of central tendency and dispersion.

Regarding the inferential analysis, in contrast to statistical hypotheses about AKI, antibiotic use, and other associated factors, the comparison of proportions test was used according to the assumptions evaluated using the chi-square test or Fisher’s exact test. In the case of a quantitative variable, normality tests were performed, so the Mann–Whitney U test and the Student’s *t*-test were performed. Values of *p* < 0.05 were considered a statistically significant association. A Poisson regression with robust variance was used. The forecast models of interest were obtained using generalized linear models.

### 2.4. Ethical Information

The protocol was submitted for evaluation to the research committee of the Faculty of Human Medicine at the Universidad Nacional de Ucayali. The present protocol respected the norms contained in the Declaration of Helsinki. The data obtained were used only for research purposes, preserving the anonymity of the participants and eliminating data susceptible to identification. The database of the present study will be eliminated two years after its publication. Finally, the authors declare that they have no conflicts of interest in the conduct of this research.

## 3. Results

The final study population consisted of 349 patients admitted to the intensive care unit of the Hospital II EsSalud Pucallpa; 73 patients with incomplete data and 11 patients with CKD were excluded (Figure 1).

The median age was 64 years (RIQ: 55–71), 65.62% were male, and the median creatinine was 0.76 (RIQ: 0.58–0.98). Regarding anthropometric measurements, the median weight was 75 kg (RIQ: 67–83.5), median height 1.69 m (1.65–1.73), and median BMI was 26.28 kg/m^2^ (RIQ: 24.14–29.06). Regarding clinical characteristics, the median time of illness was 8 days (RIQ: 7–13), the median time of hospitalization was 5 days (RIQ: 2–10), hypertension was the most frequent comorbidity with 42.24%, the most frequent complication was mechanical ventilation with 42.41%, azithromycin was used in 16.91% and vancomycin was used in 14.61%. Regarding clinical outcomes, 14.61% required renal replacement therapy, 11.27% had a successful recovery and 78.22% died (Table 1).

A total of 10.89% of patients infected with COVID-19 developed AKI. In Bivariate Analysis, the factors that were associated with AKI were BMI (*p* = 0.0020), time of illness (*p* < 0.001), days of hospitalization (*p* < 0.001), presence of shock (*p* < 0.001), sepsis (*p* < 0.001), mechanical ventilation (*p* < 0.001), venous thromboembolism (*p* = 0.010), vancomycin use (*p* < 0.001), and azithromycin (*p* = 0.003) (Table 2).

In linear regression, the factors that were positively associated with the development of AKI were BMI (cPR:1.11; 95% CI:1.05–1.17), days of hospitalization (cPR: 1.04; 95% CI: 1.03–1.04), presence of shock (cPR: 18.41; 95% CI: 10.45–32.47), sepsis (cPR: 24.34; 95% CI: 12.89–45.95), mechanical ventilation (cPR: 15.84; 95% CI: 4.96–50.62), use of vancomycin (cPR: 21.91; 95% CI: 10.64–45.11), and azithromycin (cPR: 2.55; 95% CI: 1.39–4.70). The factor negatively associated with the development of AKI was time of illness (cPR:0.81; 95% CI: 0.72–0.91). In the adjusted regression, the factors positively associated were sepsis (aPR: 2.86; 95% CI: 1.26–6.43), shock (aPR:2.49; 95% CI: 1.28–4.86), mechanical ventilation (aPR:9.11; 95% CI: 1.23–67.57), and use of vancomycin (aPR: 3.15; 95% CI: 1.19–8.27) (Table 3).

## 4. Discussion

In the population studied, the proportion of individuals with COVID-19 who developed AKI increased for each point increase in BMI. Many studies have demonstrated an increased risk of mortality in patients with COVID-19 and elevated BMI [20,21,22].

Martín del Campo et al. found that a BMI classified as morbidly obese was significantly associated with an increased risk of AKI (OR: 2.70; 95% CI: 1.01–7.25), and a higher BMI was associated with higher mortality (OR: 2.70; 95% CI: 1.01–7.25) [23].

Many mechanisms could explain the increased risk of AKI due to the point increase in BMI; however, dysregulation of fatty acid and carbohydrate metabolism, the presence of comorbidities, oxidative stress, and inflammation appear to be the most important factors associated with the development of AKI in this condition [24].

Between 5% and 10% of patients infected with SARS-CoV-2 require ICU admission, and up to 67% develop shock [25]. Shock has been implicated as the leading cause of death in 7% of COVID-19 cases and as a contributing factor in an additional 33% [25]. Four types of shock (distributive, cardiogenic, obstructive, and hypovolemic shock) have been observed in patients with COVID-19 [26].

In the present study, we report that the proportion of individuals with COVID-19 who developed AKI was increased in patients who presented with shock, compared to those who did not develop AKI. Xiaoyang et al. reported that patients with AKI, compared to those who did not develop AKI, had a higher SOFA score on admission (4.5 ± 2.1 vs. 2.8 ± 1.4, *p* = 0.002) and a higher presence of shock (47.6 vs. 25.3%, *p* = 0.042) [27].

The presence of sepsis increased the probability of developing AKI. Bezerra et al. found that the development of sepsis associated with AKI was more frequent in people with AKI KDIGO 3, compared to people with KDIGO 1. They also reported that the most frequent etiology of AKI (84.2%) was associated with sepsis [28].

Several factors may contribute to the development of sepsis associated with AKI, including local inflammation, microvascular changes, and hemodynamic changes [29,30]. In addition, the SOFA score indicates the level of organ dysfunction and severity of illness [31], which explains the association of shock and sepsis with AKI in the present study.

Mechanical organ support has always been a mainstay of intensive care, especially in the use of mechanical ventilation. Among the more than 70 million people infected worldwide with SARS-CoV-2, many have required mechanical ventilation.

The proportion of individuals with COVID-19 who developed AKI was increased in patients who had mechanical ventilation, compared to those who did not develop AKI. Similar results were reported by Almeida, who reported that AKI in patients with COVID-19 was found to be associated with mechanical ventilation (OR:7.29; 95% CI: 3.01–17.67) [32].

As previously explained, COVID-19 can cause hemodynamic instability with low tissue perfusion and require mechanical ventilation and vasoactive amines [33]. Many other authors have shown that mechanical ventilation support in acute respiratory failure was independently associated with AKI in patients with COVID-19 [33,34].

The development of AKI was increased in patients who were treated with vancomycin or azithromycin compared to those who did not receive it. This is consistent with that reported by Lima et al., who estimated that the risk of developing AKI is three times higher in patients who received antibiotic therapy compared to those who did not (RR: 3.03; 95% CI: 1.64–5.62) [28]. In addition, recent studies have reported that in Latin America, particularly in Peru, 99% of hospitalized patients with COVID-19 received antibiotics unjustifiably [35,36]. In their study, Mousavi Movahed SM et al. reported that antibiotics and acute kidney injury were significantly associated with severity and death in patients hospitalized for COVID-19 [37].

Potentially nephrotoxic drugs that are commonly used in critically ill patients are antimicrobial agents (antibiotics, antiviral and antifungal therapy) [38]. Nephrotoxicity of some antimicrobial drugs is a common problem. Of these, the most nephrotoxic are vancomycin, aminoglycosides, and polymyxins, which cause acute tubular necrosis and apoptosis, depending on the dose, among other factors. Macrolides such as azithromycin cause acute tubulointerstitial nephritis [39]. One possible explanation for vancomycin-induced AKI and tubular cell apoptosis is based on DNA methylation through activation of the methyl-CpG-binding domain protein 2 (MBD2) [40].

According to studies, the use of antibiotics for the treatment of COVID-19 results in increased hospital mortality [41], acute organic lesions [42,43,44], and antimicrobial resistance, which makes it difficult to identify the priority of treatment groups and to improve the care of common bacterial infections in the region [45]. However, antibiotics have reduced the burden of many infectious diseases, and their misuse has contributed to significant adverse effects affecting health and the economy [46]. The management of COVID-19 patients in Asia and Europe showed that 63.9% of those hospitalized received antibiotic treatment, and 62.4% received antiviral treatment (interferon, ribavirin, lopinavir-ritonavir) [47]. In this study, the drugs used in the treatment of COVID-19 were aminoglycosides, vancomycin, ivermectin, azithromycin, tociluzumab, and corticosteroids.

For the treatment of patients with acute kidney injury and COVID-19, it is necessary to provide active antiviral and oxygen therapies, respiratory support, circulation monitoring, and nutritional support therapy. Among the antiviral drugs, we have lopinavir, ritonavir, abidor, and shufeng. In addition, drugs that inhibit cytokine storms are used, such as tocilizumab, sturmuzumab, etanercept, certolizumab, anakinra, and cyclosporine A [48]. In a study of 854 patients, the most common drugs prescribed were diphenhydramine and azithromycin in 60.5% and 52.8%, respectively. Furthermore, patients who had received antibiotics against co-infections such as linezolid, vancomycin, carbapenem, cephalosporin, and piperacillin/tazobactam developed significantly higher acute kidney injury [37].

Peru is one of the countries where there is a high consumption and inappropriate prescription of antibiotics; therefore, it is important to propose programs and strategies to control the appropriate use of these drugs [49].

The present study has limitations. First, because this is a cross-sectional analytic study, we cannot determine causal relationships between exposures and AKI. Second, although we adjusted for potential confounders, there may be unmeasured potential confounders. Third, we relied exclusively on medical records (digital and physical) as a source of data, including AKI identification. However, the amount of data validation, cleaning, auditing, and quality control was substantial. Fourth, since this is a single-center, census-type survey and no randomization of participants was performed, the results of the investigation cannot be extrapolated to the total population of Ucayali; however, the results provide us with an approach for the use of antibiotics and their impact on the development of AKI during the first waves of the COVID-19 pandemic in the Ucayali region. Fifth, vancomycin toxicity may or may not be directly related to COVID-19 and influence the development of acute kidney injury.

## 5. Conclusions

As previously described [33,34,35,36], the inappropriate use of antibiotics for COVID-19 has been associated with multiple negative outcomes and consequences. The current study concludes that the development of AKI among hospitalized patients was associated with vancomycin administration. In addition, an association was found with the use of mechanical ventilation, high BMI, and the presence of complications such as sepsis or shock.

## 6. Recommendations

The present study provides highly relevant information about the association between antibiotic use and the development of acute kidney injury in patients infected with SARS-CoV-2. In addition, approximately 76% of the patients who developed AKI were also septic. This raises an interesting question that would lead to the development of a similar study in ICU/ventilated/septic patients who were not positive for COVID-19.

## Figures and Tables

**Figure 1 jcm-11-04493-f001:**
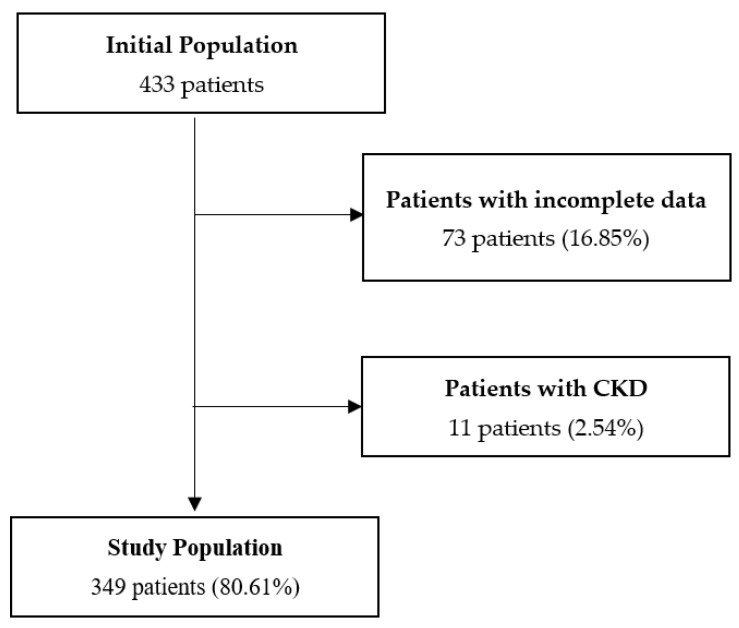
Patient selection flowchart.

**Table 1 jcm-11-04493-t001:** Sociodemographic and clinical characteristics of patients infected with SARS-CoV-2 treated at Hospital EsSalud II Ucayali.

Variable	*n* = 349	%
Sex		
Female	120	34.38
Male	229	65.62
Age	64	(55–71) *
Creatinine ^†^	0.76	(0.58–0.98) *
Weight ^†^	75	(67–83.5) *
Size ^†^	1.69	(1.65–1.73) *
BMI ^†^	26.28	(24.14–29.06) *
Time of illness (days) ^†^	8	(7–13) *
Days of hospitalization ^†^	5	(2–10) *
Comorbidities ^†^		
Hypertension	147	42.24
Diabetes Mellitus	93	26.72
Neoplasms	1	0.29
Cardiovascular disease	31	8.91
Pulmonary Disease	16	4.60
Diabetes + Hypertension	63	18.05
Complications of COVID-19		
Sepsis	36	10.32
Shock	33	9.46
Mechanical ventilation ^¥^	148	42.41
Venous thromboembolism	5	1.43
Medications ^†^		
Aminoglycosides	14	4.01
Vancomycin	51	14.61
Ivermectin	40	11.46
Azithromycin	59	16.91
Tociluzumab	9	2.58
Corticosteroids	326	93.41
Ivermectin + Azithromycin	27	7.74
Ivermectin + Azithromycin + Corticosteroids	23	6.59
Azithromycin + Corticosteroids	55	15.76
Renal Replacement Therapy ^†^	51	14.61
Recovery ^†^	39	11.27
Death ^†^	273	78.22

^†^ Some variables may add up to less than 354 because of missing data. ^¥^ Includes invasive and noninvasive mechanical ventilation. * Median (Interquartile ranges 25–75%).

**Table 2 jcm-11-04493-t002:** Bivariate analysis between AKI, antibiotic use and associated factors in patients infected with SARS-CoV-2 treated at Hospital EsSalud II Ucayali.

Variables	Acute Renal Failure		*p*
	No *n* (%)	Yes *n* (%)	
Sex			
Female	109 (90.83)	11 (9.17)	0.455 *
Male	202 (88.21)	27 (11.79)	
Age	64 (55–72) ^¥^	59 (55–66) ^¥^	0.034 ^†^
Weight	75 (67–82) ^¥^	76 (68–89) ^¥^	**0.289 ^†^ **
Size	1.69 (1.65–1.73) ^¥^	1.65 (1.59–1.70) ^¥^	**0.001 ^†^ **
BMI	26.07 (23.74–28.73) ^¥^	27.69 (25.71–31.23) ^¥^	**0.002 ^†^ **
Time of illness (days)	9 (7–13) ^¥^	7 (5–7) ^¥^	**<0.001 ^†^ **
Days of hospitalization	4 (2–8) ^¥^	28 (19–43) ^¥^	**<0.001 ^†^ **
Comorbidities			
Hypertension	132 (89.80)	15 (10.20)	0.714 *
Diabetes Mellitus	85 (91.40)	8 (8.60)	0.403 *
Neoplasms	--	---	---
Cardiovascular disease	27 (87.10)	4 (12.90)	0.761 **
Pulmonary Disease	15 (93.75)	1 (6.25)	1.000 **
Diabetes + Hypertension	59 (93.65)	4(6.35)	0.201 *
Complications of COVID-19			
Sepsis	8 (22.22)	28 (77.78)	**<0.001 ****
Shock	8 (24.24)	25 (75.76)	**<0.001 ****
Mechanical ventilation	114 (76.51)	35 (23.49)	**<0.001 ***
Venous thromboembolism	2 (40.00)	3 (60.00)	**0.010 ****
Drugs used in the treatment of COVID-19			
Aminoglycosides	13 (92.86)	1 (7.14)	1.000 **
Vancomycin	21 (41.18)	30 (58.82)	**<0.001 ***
Ivermectin	33 (82.50)	7 (17.50)	0.174 **
Azithromycin	46 (77.97)	13 (22.03)	**0.003 ***
Tociluzumab	8 (88.89)	1 (11.11)	1.000 **
Corticosteroids	293 (89.88)	33 (10.12)	0.090 **
Ivermectin + Azithromycin	20 (74.06)	7 (25.93)	0.018 **
Ivermectin + Azithromycin + Corticosteroids	18 (78.26)	5 (21.74)	0.090 **
Azithromycin + Corticosteroids	44 (80.00)	11 (20.00)	0.018 *

* Calculated with the chi-square test of independence; ** Calculated with Fisher’s exact test; ^†^ Calculated with the Mann–Whitney U test, ^¥^ Median (Interquartile ranges 25–75%).

**Table 3 jcm-11-04493-t003:** Factors associated with AKI and antibiotic use in patients infected with SARS-CoV-2 treated at Hospital EsSalud II Ucayali, simple and multiple regression analysis.

Variables	Simple Regression			Multiple Regression		
	cPR	95% CI	*p*	aPR	95% CI	*p*
Sex						
Female	Ref.	-	-	-	-	-
Male	1.29	0.66–2.50	0.459	-	-	-
Age	0.98	0.97–0.99	0.033	1.00	0.98–1.03	0.938
BMI	**1.11**	**1.05–1.17**	**<0.001**	1.00	0.96–1.06	0.785
Time of illness (days)	**0.81**	**0.72–0.91**	**<0.001**	**0.92**	**0.84–1.00**	**0.078**
Days of hospitalization	**1.04**	**1.03–1.04**	**<0.001**	1.00	0.99–1.02	0.596
Comorbidities						
Hypertension	0.89	0.48–1.65	0.715	-	-	-
Diabetes Mellitus	0.73	0.35–1.54	0.409	-	-	-
Cardiovascular disease	1.20	0.46–3.17	0.709	-	-	-
Pulmonary Disease	0.56	0.08–3.84	0.556	-	-	-
Diabetes + Hypertension	0.53	0.19–1.45	0.219	-	-	-
Complications of COVID-19						
Sepsis	24.34	12.89–45.95	<0.001	**2.86**	**1.26–6.43**	**0.012**
Shock	**18.41**	**10.45–32.47**	**<0.001**	**2.49**	**1.28–4.86**	**0.007**
Mechanical ventilation	**15.84**	**4.96–50.62**	**<0.001**	**9.11**	**1.23–67.57**	**0.031**
Drugs used in the treatment of COVID-19 *						
Aminoglycosides	0.65	0.95–4.39	0.656	-	-	-
Vancomycin	**21.91**	**10.64–45.11**	**<0.001**	**3.15**	**1.19–8.27**	**0.020**
Ivermectin	1.74	0.82–3.70	0.147	-	-	-
Azithromycin	**2.55**	**1.39–4.70**	**0.003**	3.10	0.23–42.31	0.396
Corticosteroids	0.47	0.20–1.08	0.075	-	-	-
Ivermectin + Azithromycin	2.69	1.30–5.54	0.007	1.97	0.74–5.25	0.177
Ivermectin + Azithromycin + Corticosteroids	2.15	0.93–4.98	0.075	-	-	-
Azithromycin + Corticosteroids	2.18	1.15–4.13	0.017	0.39	0.03–5.02	0.475

Values obtained with generalized linear models of the Poisson family with robust variance; cPR: crude Prevalence Ratio; aPR: adjusted Prevalence Ratio; 95% CI: 95% confidence interval. * Non-standardized doses.

## Data Availability

The data are not publicly available due to their containing information that could compromise the privacy of research participants. Permits were obtained from the EsSalud II Hospital in Pucallpa for its execution, as well as from the head of the Nephrology Service.

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
