# Peer review of "Association between the Use of Antibiotics and the Development of Acute Renal Injury in Patients Hospitalized for COVID-19 in a Hospital in the Peruvian Amazon"

_jcm, 2022, doi:10.3390/jcm11154493_

Round 1

Reviewer 1 Report

This study aims to investigate the association between the use of antibiotics and the development of acute renal injury in 24 patients infected with SARS-CoV-2 in patients treated at Hospital II EsSalud de Ucayali 2021.

The authors showed that the development of Acute Renal Failure among hospitalized patients was  associated with the administration of antibiotics, with the use of mechanical ventilation, high BMI, and with the presence of complications such as sepsis or shock..

MAJOR COMMENTS

-       In the text before indicating the acronym, use the definition in full. For example in line 29 is written ARF without the definition. In line 30 also add the BMI among the factors positively associated with the development of ARF.

-       Introduction. I suggest providing a more detailed presentation of pathophysiology of acute renal failure in patients with COVID-19, in this regard, discuss this interesting paper (doi: 10.1007/s15010-021-01706-6; PMID  34611792). In addition , it will be for the benefit of the reader if the author mentions the management of ARF in these patients and underlines that ARF is a marker of disease severity and a negative prognostic factor for survival (doi: 10.1016/S2213-2600(20)30229-0; PMID 32416769).

-       Being the association between acute kidney injury (AKI) and the use of non-steroidal anti-inflammatory drugs (NSAIDs) well established, how many patients were being treated with these drugs? Authors should add in the text information about possible causes of acute renal failure as shown in the table 3.

-       Discussion. Between the limits add that it is a monocentric study.

Author Response

Thank you very much for your reviews and comments on this article.

  1. We have corrected the acronyms using the full definition. Acute kidney injury (AKI), body mass index (BMI).
  2. We have organized the introduction by presenting in more detail the pathophysiology of acute renal failure in patients with COVID-19, in addition to the articles recommended: (doi: 10.1007/s15010-021-01706-6; PMID 34611792) y (doi: 10.1016/S2213-2600(20)30229-0; PMID 32416769).
  3. We have added information in the introduction section on the possible causes of acute renal failure, as shown in Table 3.
  4. We have indicated a single-center study as the study limit.
  5. Based on your recommendations we organized the best conclusion.

Reviewer 2 Report

-        Line 61 - The reference #6 that evaluated incidence of AKI throughout the world is assessing COVID-19 patients only in the first wave of the pandemic – this section of introduction should be changed accordingly. Furthermore, new references that evaluate AKI in the other pandemic waves should be added.

-        How was sample size determined? Moreover, more information regarding detailed steps of the patient enrolment, as well as informed consent collection is needed.

-        Table 3 is written poorly, with models that should be present in the multiple regression analysis are not clear. Furthermore, all of the Results lack flow, with many information and statistical tests added without clear direction.

-        Line 61 - The reference #6 that evaluated incidence of AKI throughout the world is assessing COVID-19 patients only in the first wave of the pandemic – this section of introduction should be changed accordingly. Furthermore, new references that evaluate AKI in the other pandemic waves should be added.

-        How was sample size determined? Moreover, more information regarding detailed steps of the patient enrolment, as well as informed consent collection is needed.

-        Table 3 is written poorly, with models that should be present in the multiple regression analysis are not clear and emphasized. Furthermore, all of the Results lack flow, with many information and statistical tests added without clear direction.

-        In Discussion, clinical implications that can be derived from the results of this study should be expanded.

Author Response

Thank you very much for your reviews and comments on this article.

  1. We have added information based on new references evaluating AKI in the other waves of the pandemic.
  2. We have organized and added information on the sample: The sample size was carried out by a non-probabilistic census-type sampling, based on the medical records provided by the patients provided by the Hospital EsSalud II of Pucallpa. Therefore, an informed consent form was not used for each patient.
  3. We organized the wording of Table 3, emphasizing the information on factors associated with ARI and antibiotic use in patients infected with SARS-CoV-2.

Reviewer 3 Report

The manuscript has interesting content. However, the originality must be improved and presentation must be carefully edited for appropiateness. 

a) The entire manuscript should be revised to avoid typo, grammar and punctuation mistakes.

b) The terms AKI and/or ARF (or ARI) should be clearly explained and used with clarity.

c) Introduction should be updated, considering some new facts of pandemic. 

d) Some graphical representation is desirable about the relationship between factors (including use of antibiotics) and AKI. 

e) The points of originality should be highlighted. It seems to be that originality is just confered by the population or region where the study was carried out.

f) The identification of the pair of antibiotics linked to ARF should be clearly analyzed. Many recent works are related to the role of specific antibiotics in the COVID pathophysiology, including changes in Kidney function. In the abstract just vancomicin is mentioned, while other factors are unrelated to antibiotics therapy.

g) Clear association and differences should be with your recent work: Copaja-Corzo, C., Hueda-Zavaleta, M., Benites-Zapata, V. A., & Rodriguez-Morales, A. J. (2021). Antibiotic use and fatal outcomes among critically ill patients with covid-19 in Tacna, Peru. Antibiotics10(8), 959.

h) Several recent works should be considered to enrich discussion. Examples:

Mousavi Movahed, S. M., Akhavizadegan, H., Dolatkhani, F., Nejadghaderi, S. A., Aghajani, F., Faghir Gangi, M., ... & Ghasemi, H. (2021). Different incidences of acute kidney injury (AKI) and outcomes in COVID‐19 patients with and without non‐azithromycin antibiotics: A retrospective study. Journal of medical virology93(7), 4411-4419.

Goncalves Mendes Neto, A., Lo, K. B., Wattoo, A., Salacup, G., Pelayo, J., DeJoy III, R., ... & Azmaiparashvili, Z. (2021). Bacterial infections and patterns of antibiotic use in patients with COVID‐19. Journal of medical virology93(3), 1489-1495.

Zeshan, B., Karobari, M. I., Afzal, N., Siddiq, A., Basha, S., Basheer, S. N., ... & Noorani, T. Y. (2021). The usage of antibiotics by COVID-19 patients with comorbidities: the risk of increased antimicrobial resistance. Antibiotics11(1), 35.

Xia, P., Wen, Y., Duan, Y., Su, H., Cao, W., Xiao, M., ... & Li, X. (2020). Clinicopathological features and outcomes of acute kidney injury in critically ill COVID-19 with prolonged disease course: a retrospective cohort. Journal of the American Society of Nephrology31(9), 2205-2221.

Ramírez-Lozada, T., Loranca-García, M. C., Fuentes-Venado, C. E., Rodríguez-Cerdeira, C., Ocharan-Hernández, E., Soriano-Ursúa, M. A., ... & Martínez-Herrera, E. (2022). Does the Fetus Limit Antibiotic Treatment in Pregnant Patients with COVID-19?. Antibiotics11(2), 252.

Chu, K. H., Tsang, W. K., Tang, C. S., Lam, M. F., Lai, F. M., To, K. F., ... & Lai, K. N. (2005). Acute renal impairment in coronavirus-associated severe acute respiratory syndrome. Kidney international67(2), 698-705.

Author Response

Thank you very much for your reviews and comments on this article.

  1. The manuscript has been reviewed by experts in the English language to avoid typographical, grammatical, and punctuation errors.
  2. We have organized the IRA and/or FRA (or IRA) terms so that they can be used correctly throughout the manuscript.
  3. We have updated the introduction, organizing some epidemiological and physiological concepts and processes of acute kidney injury and COVID-19. Updated and high-impact articles have been added.
  4. For the present article, we were not able to design a graphical representation scheme on the relationship between factors (including antibiotic use) and AKI. However, the presentation of tables and figures details the information.
  5. We have better organized the information supporting the originality of the work and also specifying some limitations regarding the approach of the study in the population of the region, since it was the most affected and has had an impact on the health sector.
  6. We have added the interesting articles recommended in the introduction and discussion sections.

Round 2

Reviewer 1 Report

Authors answered all comments and suggestions.

Author Response

Thanks for you comments

Reviewer 3 Report

Please highlight the particular data from the analysis in the peruvian amazonas. Vancomicin is the only one antibiotic related to Renal failure as is sentenced in abstract? What about Azithromicine? Some other data from your tables could be carefully discussed.  

The conclusions should to be specific about the data regarding antibiotics use in your country. 

Author Response

Dear reviewer.

Thank you for your recommendations to improve the quality of this article.

We have made an update of the recently published articles according to the topic addressed.

The detailed antibiotics used for the treatment of COVID-19 have been added in the abstract.

In the conclusion, the recommendations that may result from the use of these antibiotics.

In the discussion section, the drugs used in the treatment in the Peruvian Amazon area were added, comparing with other recent studies.

The recommended articles on the use of antibiotics for acute kidney injury were added, which the authors consider an improvement of the evidence to be able to discuss their results.